# Molecular folding governs switchable singlet oxygen photoproduction in porphyrin-decorated bistable rotaxanes
Jan Riebe [1], Benedikt Bädorf [2], Sarah Löffelsender[2], Matias E. Gutierrez Suburu[3],
María Belén Rivas Aiello[3], Cristian A. Strassert[3], Stefan Grimme [2] ✉ & Jochen Niemeyer [1] ✉

Rotaxanes are mechanically interlocked molecules where a ring (macrocycle) is threaded onto a linear molecule (thread). The position of the macrocycle on different stations on the thread can be controlled in response to external stimuli, making rotaxanes applicable as molecular switches. Here we show that bistable rotaxanes based on the combination of a Zn(II) tetraphenylporphyrin photosensitizer, attached to the macrocycle, and a black-hole-quencher, attached to the thread, are capable of singlet oxygen production which can be switched on/off by the addition of base/acid. However, we found that only a sufficiently long linker between both stations on the thread enabled switchability, and that the direction of switching was inversed with regard to the original design. This unexpected behavior was attributed to intramolecular folding of the rotaxanes, as indicated by extensive theoretical calculations. This evidences the importance to take into account the conformational flexibility of large molecular structures when designing functional switchable systems.

Molecular switches are molecules that exist in different states which can be reversibly addressed. Most switches feature two stable switching states that can be selectively addressed by an external stimulus[1–4]. The change in molecular properties which results from the switching process can be used to deliver a function, which has allowed the construction of a plethora of functional materials based on molecular switches, such as molecular muscles and elevators[5,6], nanovalves and other systems for controlled release[7,8], materials with switchable surface properties[9], switchable catalysts[10–14] and materials with switchable photophysical properties[15–17].

Mechanically interlocked molecules, such as rotaxanes (and, to a lesser extent, catenanes) provide an excellent platform for the construction of functional molecular switches[18–22]. In a bistable rotaxane, featuring a macrocycle that encircles a thread with two binding stations, the preferred position of the macrocycle is determined by its relative affinity to the binding stations. However, when an external stimulus can be employed to modify one station, this can invert the relative affinities of the two binding stations. This infers a large amplitude motion of the macrocycle along the thread, which moves away from the (modified) first station and binds to the second station. Reversing this process leads to re-binding of the macrocycle at the first station. Like molecular switches in general, switchable bistable rotaxanes have been based on a number of switching mechanisms, most

commonly (but not limited to) changes in temperature, irradiation with light, application of an electric potential, addition/removal of other chemicals or a change in pH. One of the most widely used types of bistable rotaxanes features a crown-ether-based macrocycle and a thread with an amine/ammonium station plus a triazolium station (see Fig. 1)[23,24]. Deprotonation of the ammonium station, which is the preferred binding station when protonated, leads to relocation of the macrocycle towards the triazolium station (and vice versa, with the respective co-conformer being the predominant one by >99%, making the switching effectively binary[25]). This motion has been used to shield/reveal the amine/ammonium station by the macrocycle (e.g., for the design of switchable catalysts[26–29], see Fig. 1a), to change the co-conformation between stretched/contracted conformers (e.g., for the design of molecular muscles[30,31], see Fig. 1b) or to change the distance and/or environment of chromophores attached to the sub-components (e.g., for the design of molecules with tunable photophysical properties, such as fluorescence, see Fig. 1c)[32–38]. However, it was also realized that the prediction of the molecular conformations in each switching state (and with this, the function in each state) is not trivial. For example, Leigh and coworkers found that in the protonated state, their rotaxane (see Fig. 1a) can adopt a folded conformation that brings both triazolium units in close proximity. This allows the use of the protonated rotaxane as a catalyst

[1]Faculty of Chemistry (Organic Chemistry) and Center for Nanointegration Duisburg-Essen (CENIDE), University of Duisburg-Essen, Universitätsstrasse 7, 45141 Essen, Germany. [2]Mulliken Center for Theoretical Chemistry, Rheinische Friedrich-Wilhelms-Universität Bonn, Beringstrasse 4, 53115 Bonn, Germany. [3]Institut für Anorganische und Analytische Chemie, CeNTech, CiMIC, SoN, Universität Münster, Heisenbergstr. 11, 48149 Münster, Germany. ✉ e-mail: grimme@thch.uni-bonn.de; jochen.niemeyer@uni-due.de

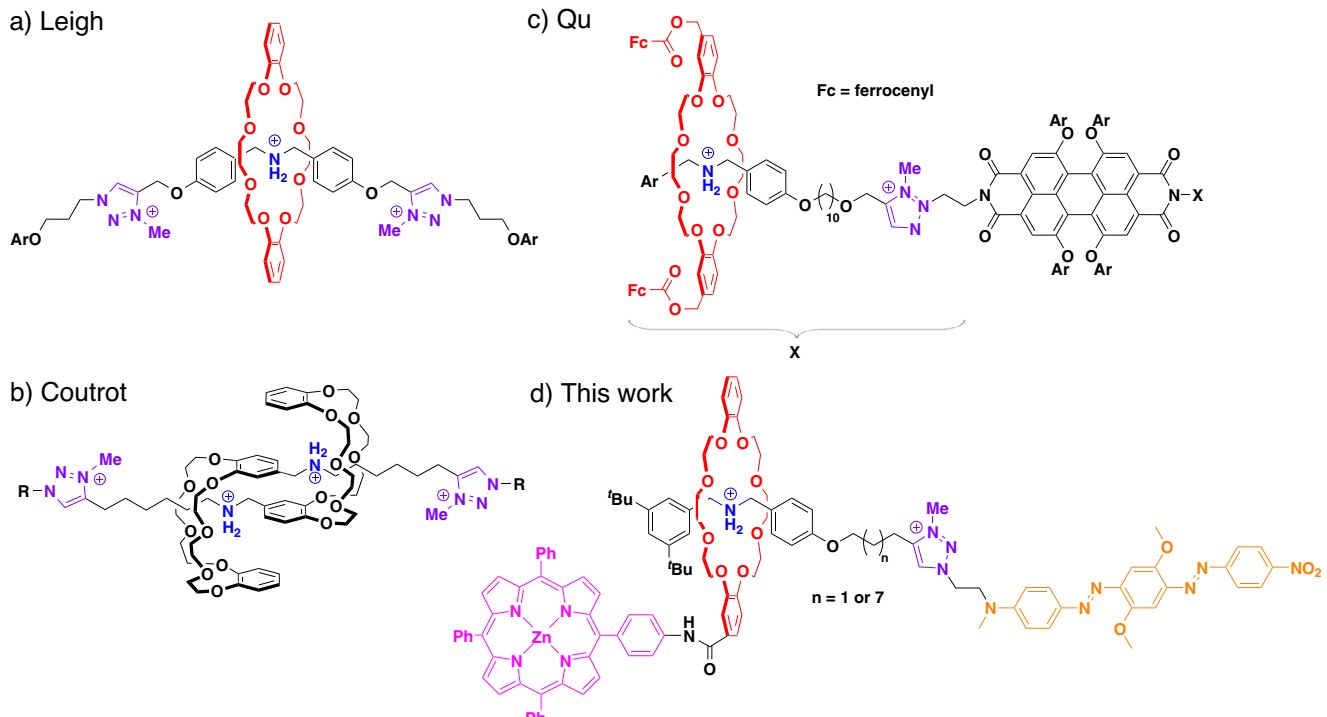

**Fig. 1 | Selected molecular switches based on bistable rotaxanes with ammonium/amine plus triazolium stations. a** Switchable catalysts[29], **b** molecular muscles[31], **c** switchable chromophors[34], **d** switchable systems for $^1O_2$ photoproduction (this work).

via halide-abstraction, enabled by chelate-type binding of halides by the triazolium units[26].

The development of switchable photosensitizers for singlet dioxygen ($^1O_2$) photoproduction (for simplicity, referred to hereafter as: singlet oxygen production) is an important field of application for functional materials. Singlet oxygen can be used for medical purposes, such as photodynamic therapy[39,40] or disinfection[41,42], but also as a reagent for organic synthesis[43–46]. However, due to its high reactivity and cell toxicity, there is an ongoing interest in smart materials that produce singlet oxygen only after switching to an on-state, while the singlet oxygen production is suppressed in an off state[47–52]. Various approaches have been used to design such systems, especially in the context of photodynamic therapy. Most systems rely on the quenching of photosensitizers in the off-state (suppressing singlet oxygen production), which can be realized via different mechanisms. Importantly, this includes systems that can be activated in a non-reversible or in a reversible form. For example, non-reversible activation has been realized by integration of photosensitizers into hybrid organic/inorganic materials based on layered double hydroxides or calcium phosphate (leading to quenching), which release the photosensitizer in its non-quenched on-state at low pH[53,54]. In addition, photosensitizer-quencher conjugates with cleavable linkers have shown to undergo fragmentation in a redox-dependent or enzyme-mediated fashion, leading to a non-reversible activation of singlet oxygen production[55–60]. Systems that, in principle, allow for reversible activation include polymers that feature photosensitizer-substituents, allowing for an on/off-switchable singlet oxygen production based on pH-dependent conformational changes of the polymer-backbone[61,62]. Photosensitizers or photosensitizer dyads with Lewis-basic substituents have been used to enable the on-switching of singlet oxygen production at low pH, based on the suppression of photoinduced electron transfer and/or modulation of acceptor/donor energy levels[63–65]. In a related approach, photosensitizers conjugated with diarylethene-photoswitches make use of the UV/Vis-absorption of the closed photoswitch, which leads to a quenching of the photosensitizers (off-state) and vice versa[66,67]. Photosensitizers and quenchers conjugated with complementary DNA strands have been used to achieve on/off-switching of singlet oxygen production by

reversible DNA-hybridization[68]. Finally, rotaxanes have also successfully been used to construct systems for biomedical applications in general, and for the design of switchable photosensitizers specifically[69–71]: Huang and coworkers developed a pH-switchable pseudorotaxane based on an AIEgen-based thread (AIE: aggregation-induced emission) and a pillar[5]arene macrocycle that undergoes a pH-dependent co-conformational change, leading to a switchable singlet oxygen production[72]. Lin and coworkers reported a pH-switchable [1]rotaxane-based on a crown-ether macrocycle and a multifunctional thread containing amine/ammonium and triazolium stations, plus a diarylethylene-photoswitch and an AIEgen photosensitizer. This system undergoes dual switching of photochromic behavior and singlet oxygen production controlled by pH and light[73]. Shinmori and coworkers synthesized a [2]rotaxane featuring a gold-nanoparticle-stopper (which acts as a quencher), a crown-ether macrocycle substituted with a porphyrin photosensitizer and a single amine/ammonium station on the thread[74]. This system shows pH-dependent switching, although reversibility is low, and the singlet oxygen quantum yields of the switching states are quite similar.

Based on this precedence, we envisaged that bistable [2]rotaxanes containing a macrocycle substituted with a suitable photosensitizer, and a thread functionalized with a suitable quencher, should allow for the construction of a reversibly switchable system for singlet oxygen production. Ideally, singlet oxygen production in the off-mode should be fully suppressed, to avoid undesired side-effects in a possible application. To this end, we designed rotaxanes **1a** and **1b** (see Fig. 1d), which commonly contain Zn(II) tetraphenylporphyrin[75] as a photosensitizer (attached to the macrocycle) and a black-hole-quencher (attached to one end of the tread), but differ in the length of the linker between both stations on the thread. We found that rotaxane **1a** indeed allows for an on/off-switching ($^1O_2$ quantum yield, $\Phi_\Delta$ = 15%/3%) by pH-induced translocation of the macrocycle on the thread, whereas rotaxane **1b** remains in an off-state independent of macrocycle position. Surprisingly, we discovered that the protonated state of **1a** represents the off-state of this molecular switch (and vice versa). Based on a series of control-rotaxanes and extensive theoretical analysis of the rotaxane conformations, we found that the switchable singlet oxygen production is

governed by molecular folding, which in turn is influenced by structural variations in the rotaxane-structures.

## Results and discussion
### Synthesis of the rotaxanes

The design of the pH-switchable rotaxanes for controllable singlet oxygen production was based on the following considerations: We employed Zn(II) tetraphenylporphyrin (ZnTPP) as a photosensitizer, due to its strong absorption at ca. 420 nm ($\varepsilon = 560{,}000$ cm$^{-1}$ M$^{-1}$)[76] and its high quantum yield of singlet oxygen production (($\Phi_\Delta \approx 0.7$)[77]. As a quencher, we employed the black-hole-quencher 6 (BHQ-2)[78], which features a dialkylaniline, a central dimethoxybenzene and a terminal nitrobenzene which are linked by two diazo groups. The BHQ unit shows a broad absorption band in the range of 350–650 nm is thus suitable as a FRET-acceptor (FRET: Förster resonance energy transfer) for the ZnTPP unit, which shows fluorescence emission in the range of 550–720 nm. This should suppress singlet oxygen production when both units are in close proximity.

For the combination of ZnTPP and BHQ-2, the calculated Förster-radius amounts to 2.8 nm (see SI section 1.4.2 for details). We envisaged that a sufficiently large conformational switching, which allows for moving ZnTPP and BHQ within/out of their Förster radius, should be possible by employing a bistable rotaxane with a sufficiently long linker between both stations. Here, we opted for a pH-switchable rotaxane consisting of a thread with an ammonium/amine- and a triazolium station and a dibenzo[24]crown[8] macrocycle. Attaching the ZnTPP to the macrocycle and using the BHQ as one of the stopper groups on the thread (in close proximity to the triazolium station), the rotaxane was designed to allow switching as follows: In the protonated state, the macrocycle is located around the ammonium station, sufficiently increasing the distance between ZnTPP and BHQ, resulting in $^1O_2$ production of the ZnTPP (on-state). In turn, deprotonation should locate the macrocycle at the triazolium station in closer proximity to the BHQ unit, leading to efficient FRET-based quenching and suppression of the $^1O_2$ production (off-state).

The synthesis of these functionalized rotaxanes 1a/b (differing in the length of the alkylene spacer between both stations) was realized as follows

(see Fig. 2): First, an amino-substituted ZnTPP[79] was coupled to a carboxylic-acid functionalized dibenzo[24]crown[8][80] via amide coupling to yield the ZnTPP-appended macrocycle 5. Then, the dibenzylammonium-based half-threads H-4a$^+$/H-4b$^+$ featuring a di-tert-butylphenyl-stopper on one side and an alkyne-terminated chain on the other side (undecynyl for a, pentynyl for b) were constructed by reductive amination, followed by protonation with HPF$_6$. Upon mixing of 5 with H-4a$^+$/H-4b$^+$, formation of the pseudorotaxanes was observed in a slow-exchange regime, with association constants of 61/142 L mol$^{-1}$ (see SI Figs. S19 and S20). This allowed for synthesis of the rotaxanes by stoppering of the pseudorotaxanes, which was achieved by Cu-catalyzed alkyne-azide click reaction using the BHQ-azide 5 as the coupling partner. Finally, methylation of the triazole with methyl iodide and ion-exchange with ammonium-hexafluorophosphate yielded rotaxanes H-1a$^{2+}$/H-1b$^{2+}$ in their protonated ammonium form as the bishexafluorophosphate salts (obtained in 10/9% yield over two steps from the half-threads H-4a$^+$/H-4b$^+$).

### Acid/base inducing conformational switching

The protonated rotaxanes H-1a$^{2+}$/H-1b$^{2+}$ undergo conformational switching by reversible deprotonation/reprotonation with base/acid. Change of the protonation state leads to distinct chemical shift changes in the $^1$H-NMR spectrum, in line with other amine-/triazolium-rotaxanes described earlier[24] (see Fig. 3 for 1a, see SI Fig. S14 for 1b): Upon deprotonation with NaOH, the signals of the methylene protons H-3 and H-4 (see Fig. 2 for numbering) next to the ammonium/amine group move upfield ($\Delta\delta = -1.2$ ppm) not only due to loss of the positive charge on the nitrogen atom after deprotonation, but also because of the loss of hydrogen-bonding interactions with the crown-ether's oxygen atoms. Similarly, the signal for proton H-6 on the electron-rich phenylene near the amine station experiences the expected small upfield shift of $-0.1$ ppm, due to loss of the positive charge on the ammonium. Successful relocation of the macrocycle onto the triazolium station is indicated by a drastic downfield shift of the signal of the aromatic triazolium proton H-8 by $+1.5$ ppm as well as the adjacent signal of the methylene protons H-7 by $+0.7$ ppm, caused by hydrogen-bonding to the crown-ether's oxygen atoms.

**Fig. 2 | Synthesis of the rotaxanes 1a/b.** i) Cu(MeCN)$_4$PF$_6$, CH$_2$Cl$_2$, r. t., 16 h, then methyl iodide, r. t., 24–48 hours, 10%/9% yield over two steps; ii) aq. NaOH (2 eq.), acetone, followed by removal of excess NaOH; iii) CF$_3$COOH (1 eq.). In situ acid/base switching leads to the introduction of additional cations/anions, which are not shown.

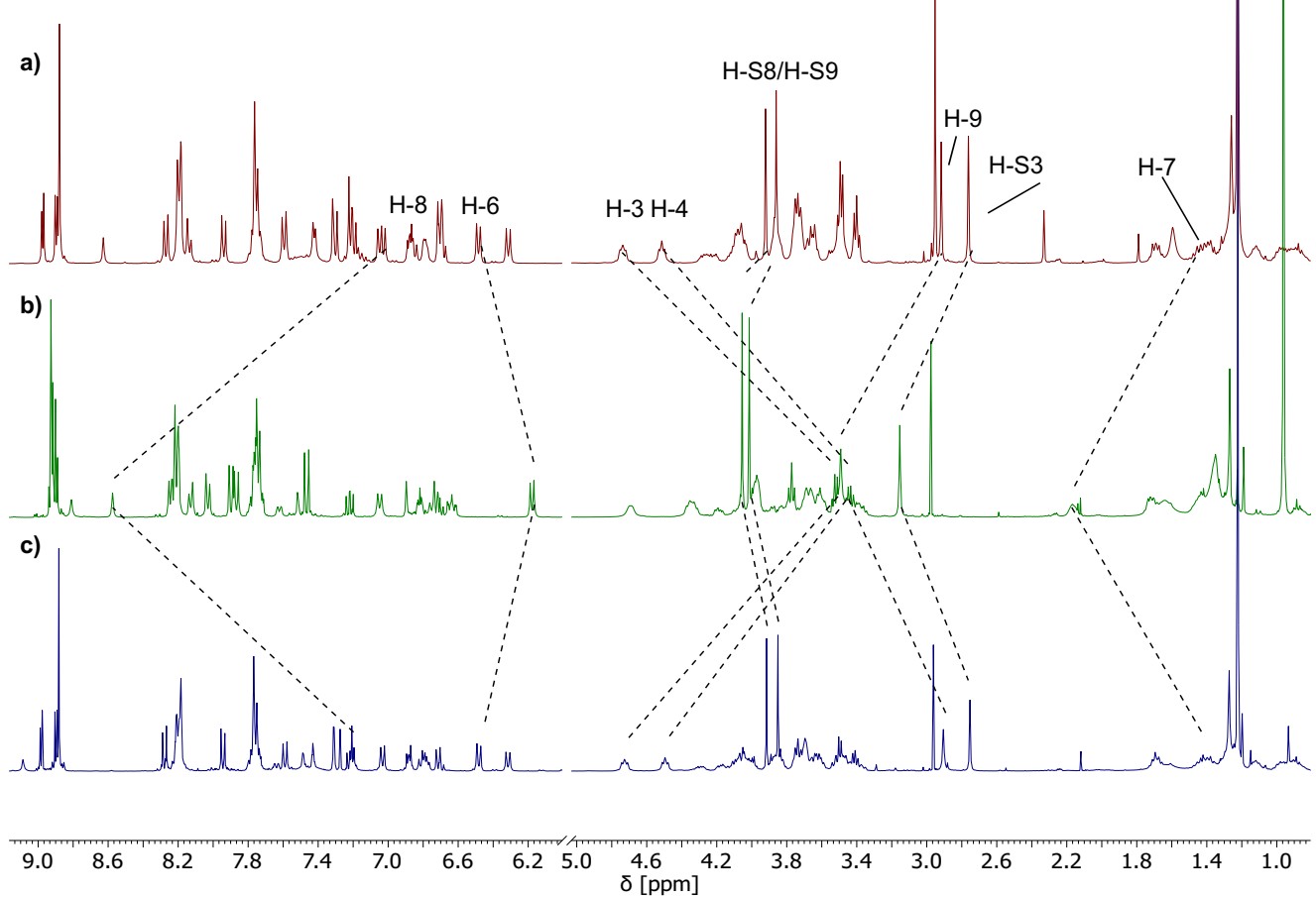

**Fig. 3 | Switching between rotaxanes H-1a²⁺ and 1a⁺ by deprotonation/reprotonation, followed by ¹H-NMR. a** Rotaxane **H-1a²⁺** (as synthesized), **b** rotaxane **1a⁺** after deprotonation. The signals at around 4.6 ppm in this spectrum could not be unambiguously assigned, but do not correspond to H-3/H-4, which can clearly be assigned. **c** Solution from **b** after addition of CF₃COOH (1 eq.) in situ. All spectra: CD₂Cl₂, 400 MHz, 298 K. For numbering of the positions, see Fig. 2.

However, some shifts were unexpected: The signals of the N-methyl group H-S3 and the methoxy groups H-S8 and H-S9 on the BHQ moiety are also shifted ($\Delta\delta$ = +0.4 ppm and +0.1 ppm, respectively), despite their large distance from the triazolium station. Furthermore, the downfield shift of the triazolium methyl group H-9 ($\Delta\delta$ = +0.6 ppm) upon deprotonation is counterintuitive, because an upfield shift would be expected[24]. Such a downfield shift for H-9 is observed in all rotaxanes (**1a/b⁺**, **2⁺**, and **3⁺**) in this study, but not with an analog of **3** possessing no ZnTPP substituent on the macrocycle (**S20-H(PF₆)₂**, see SI Fig. S17). Additionally, the switching of this rotaxane and **H-1a²⁺** was also investigated in THF-$d_8$, showing almost identical shifts of characteristic signals as in DCM-$d_2$ (for example, H-3 and H-4 move upfield by −1.3 ppm, see SI Figs. S13 and S18). Unfortunately, an attempt to prepare the non-interlocked thread for comparison was unsuccessful. In summary, these unexpected changes in chemical shift could indicate secondary interactions resulting from molecular folding of the rotaxane-structures (see theoretical investigation of the conformation space, *vide infra*).

The reverse switching can be achieved by the addition of CF₃COOH, which restores the original NMR spectrum with only minor perturbations (see Fig. 1c), probably caused by the newly introduced trifluoroacetate counter-anion.

## Investigation of ¹O₂ photoproduction

With both pH-switchable rotaxanes in hand, we investigated whether the acid/base-induced conformational change influences the ¹O₂ production upon irradiation. To follow the ¹O₂ production over time, solutions of the rotaxanes in THF were irradiated at 420 nm in the presence of excess diphenylisobenzofuran (DPBF), which acts as a ¹O₂ scavenger[81]. The decrease in DPBF-fluorescence intensity was monitored over time (see SI Figs. S21–S23) and used to calculate the singlet oxygen quantum yields (from triplicate measurements, after subtraction of the background reaction, see SI section 1.4.1 for details). For validation of this method, the $\Phi_\Delta$ of unsubstituted ZnTPP was also measured by quantification of the singlet oxygen phosphorescence, giving a near-identical result ($\Phi_\Delta$ = 0.66 ± 0.03 via ¹O₂-phosphorescence, $\Phi_\Delta$ = 0.72 ± 0.12 via DPBF method). Comparison of the singlet oxygen production of ZnTPP with that of rotaxanes in both protonation states (i.e. **H-1a²⁺** and **1a⁺**; **H-2b²⁺** and **1b⁺**) delivered the following surprising results (see Fig. 4a): In all four cases, the rotaxanes show significantly reduced ¹O₂ production in comparison to free ZnTPP, however with marked differences between the systems. For **1a**, which features the longer alkylene linker, the protonated **H-1a²⁺**-state shows only minor ¹O₂ production ($\Phi_\Delta$ < 0.03), while the deprotonated **1a⁺**-state gives a significantly higher ¹O₂ quantum yield ($\Phi_\Delta$ = 0.15 ± 0.03). Thus, the deprotonated state, which is characterized by the macrocycle being located around the triazolium station, is ca. 500% more active in ¹O₂ production, which is counterintuitive based on the original design of the molecular switch (i.e., the distance between ZnTPP and BHQ should be smaller for **1a⁺**, leading to an *off*-switching). As a comparison, rotaxane **1b** with the shorter alkylene spacer shows virtually no ¹O₂ production independent of protonation state ($\Phi_\Delta$ < 0.01 for **H-1b²⁺**, $\Phi_\Delta$ < 0.01 for **1b⁺**), although the pH-responsive translocation of the macrocycle was clearly demonstrated by ¹H-NMR spectroscopy.

**Fig. 4 | $^1O_2$ production of rotaxanes in comparison to free ZnTPP. a** $^1O_2$ production of rotaxanes **H-1a$^{2+}$** (red circles) and **1a$^+$** (green triangles); **H-1b$^{2+}$** (blue triangles) and **1b$^+$** (turquoise diamonds) in comparison to free ZnTPP (black squares) (uncertainty bars for **H-1a$^{2+}$**, **H-1b$^{2+}$** and **1b$^+$** too small to be visible); **b** $^1O_2$ production of control-rotaxanes **H-2$^{2+}$** (purple hexagons) and **2$^+$** (ochre hexagons); **H-3$^{2+}$** (blue pentagons) and **3$^+$** (orange triangle) in comparison to free ZnTPP (black squares). All values determined by decrease in DPBF fluorescence (all: THF, concentration of photosensitizers adjusted to absorbance of 0.8 at 424 nm, excitation at 420 nm; values given as average of triplicates, error bars are standard deviations).

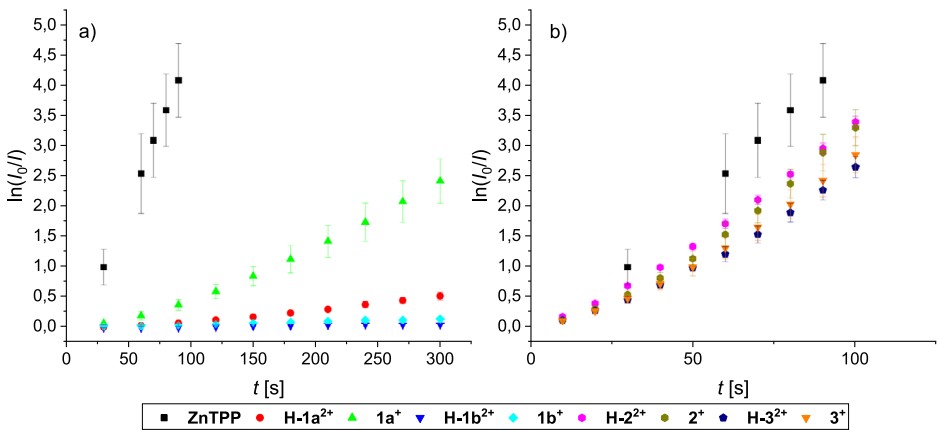

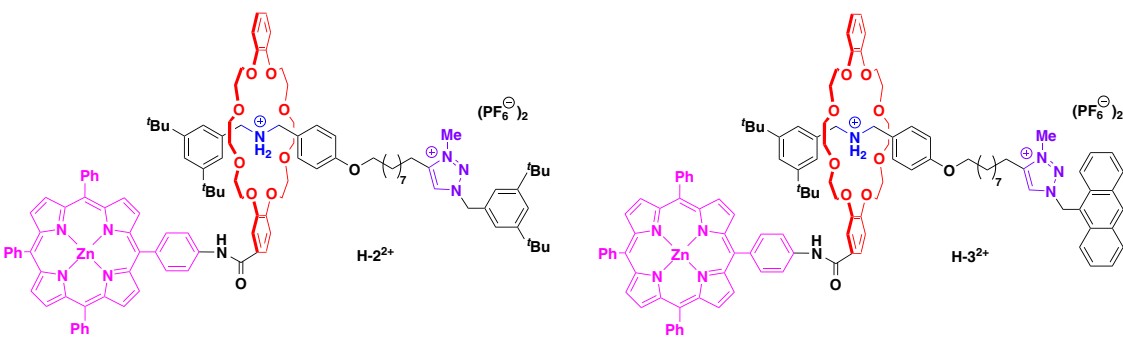

**Fig. 5 | Structural formulas of control rotaxanes.** The control rotaxanes **H-2$^{2+}$** and **H-3$^{2+}$** (only protonated states shown) were synthesized analogously to rotaxanes **H-1a$^{2+}$** and **H-1b$^{2+}$** (see SI section 1.2.2).

While these results show that bistable rotaxanes can be used to generate an *on/off*-switchable material for $^1O_2$ photosensitization, the following questions needed to be answered for an in-depth understanding:

Why is the deprotonated state **1a$^+$** the *on*-state of the molecular switch (while **H-1a$^{2+}$** is *off*)?

Why is the $^1O_2$ quantum yield low in comparison to the free ZnTPP, even in the *on*-state?

Why does the shorter spacer in **H-1b$^{2+}$**/ **1b$^+$** lead to suppression of $^1O_2$ production, independent of protonation state?

To answer these questions, we concluded that a better picture of the influence of the rotaxane architecture on the $^1O_2$ production is needed. Thus, we synthesized two control rotaxanes **2/3**, which lack the BHQ unit but instead feature a di-*tert*-butylphenyl-stopper or an anthracenyl-stopper in proximity to the triazolium station (see Fig. 5). These were synthesized in analogous fashion to **1a/b** by employing the corresponding stopper azides and were obtained in yields of 60/15% after methylation and anion-exchange. Furthermore, we attempted to synthesize analogs of **1a/b** with even longer or shorter alkylene spacers and also a variant of **1a** with an inverted thread that features the quencher in proximity to the ammonium/amine station, but these syntheses could not be realized.

With the control rotaxanes in hand, we investigated their ability to produce $^1O_2$ in both protonation states (i.e. **H-2$^{2+}$** and **2$^+$**; **H-3$^{2+}$** and **3$^+$**). We found that all four systems showed nearly identical results, with $^1O_2$ quantum yields in a range of 0.48-0.60 (see Fig. 4b). Firstly, this demonstrates that the quenching in **1a/b** is indeed affected by the BHQ unit and not by other parts of the rotaxane-architecture. Secondly, the results show that the protonation state of the ammonium/amine station alone does not influence $^1O_2$ production, but that indeed, the conformational change that places the ZnTPP and BHQ in different spatial arrangements must play a crucial role in **1a/b**. Thirdly, we found that the anthracenyl-stopper, which is

known to act as an acceptor for triplet energy transfer from Zn(II) porphyrins[82], is not suitable for modulation of $^1O_2$ photosensitization by conformational switching, showing that the nature of the BHQ quencher is crucial for excitation energy transfer.

### Fluorescence lifetimes

To obtain information about the influence of structures of rotaxanes **1a/b, 2**, and **3** on their excited state behavior, we investigated their photophysical properties. The absorption and emission spectra of all rotaxanes are governed by the ZnTPP chromophore and show only minor differences (see SI Figs. S25–S29). However, the differences observed in the $^1O_2$ production are reflected by differences in the corresponding fluorescence lifetimes (see Table 1): Here, both the ZnTPP reference and the control rotaxanes in both protonation states (i.e. **H-2$^{2+}$** and **2$^+$**; **H-3$^{2+}$** and **3$^+$**) show almost identical lifetimes of ca. 1.7–1.8 ns with monoexponential fluorescence decays. This substantiates the finding that in the absence of the BHQ unit, the rotaxane structure does not significantly influence the photophysical behavior of the ZnTPP unit, independent of conformational switching. For the BHQ-substituted rotaxanes, we observed multiexponential decay profiles, which might be indicative of different conformations that influence fluorescence lifetimes. For the rotaxane with the shorter alkylene spacer, both protonation states **H-1b$^{2+}$** and **1b$^+$** show a biexponential decay featuring a smaller component with a lifetime similar to free ZnTPP (1.76/1.57 ns, 19–29% contribution) and a dominating component with a significantly shortened lifetime (0.50/0.39 ns, 81–71% contribution), resulting in almost identical average values (0.73/0.74 ns for **H-1b$^{2+}$**/**1b$^+$**). For the rotaxane with the longer alkylene spacer (**H-1a$^{2+}$**/**1a$^+$**), we find triexponential decay kinetics, which only shows a minor component similar to free ZnTPP (1.69/1.40 ns, 5–7% contribution). The major contributions are given by two shorter lifetimes, which differ depending on the rotaxane protonation state: The protonated rotaxane **H-1a$^{2+}$** shows components at 0.87 ns (61%) and

**Table 1 | Singlet oxygen quantum yield ($\Phi_\Delta$) and amplitude-weighted average fluorescence lifetimes ($\tau_{av}$) of rotaxanes 1a/b, 2, and 3 in both protonation states in comparison to free ZnTPP[a]**

| Compound | $\Phi_\Delta$ [b] | lifetime components $\tau$ [ns] (relative amplitudes in %)[c] | Amplitude-weighted average lifetimes $\tau_{av}$ [ns] |
|---|---|---|---|
| ZnTPP | 0.72 ± 0.12 | -[d] | 1.84 ± 0.004 |
| H-1a$^{2+}$ | <0.03 | 1.69 (5), 0.87 (61), 0.30 (33) | 0.727 ± 0.016 |
| 1a$^+$ | 0.15 ± 0.03 | 1.40 (7), 0.57 (15), 0.11 (78) | 0.263 ± 0.015 |
| H-1b$^{2+}$ | <0.01 | 1.76 (19), 0.50 (81) | 0.73 ± 0.03 |
| 1b$^+$ | <0.01 | 1.57 (29), 0.39 (71) | 0.74 ± 0.05 |
| H-2$^{2+}$ | 0.60 ± 0.02 | -[d] | 1.705 ± 0.003 |
| 2$^+$ | 0.57 ± 0.06 | -[d] | 1.712 ± 0.003 |
| H-3$^{2+}$ | 0.48 ± 0.03 | -[d] | 1.712 ± 0.002 |
| 3$^+$ | 0.53 ± 0.05 | -[d] | 1.712 ± 0.002 |

[a]All measured in THF solution. [b]Concentration based on absorbance of 0.8 at 424 nm, $\Phi_\Delta$ determined from fluorescence decrease of DPBF scavenger in a linearized $\ln(I_0/I)$ vs. time plot from triplicate measurements, after subtraction of the background reaction (see section 1.4.1 of the SI for details). [c]10 µM solutions, $\lambda_{exc}$ = 407.2 nm, $\lambda_{em}$ = 605 nm, uncertainty: ±0.01 ns. [d]Only one component (given as average lifetime).

0.30 ns (33%), resulting in an average lifetime of 0.73 ns (similar to that of H-1b$^{2+}$/1b$^+$). However, for the deprotonated case 1a$^+$, a smaller component at 0.57 ns (15%) and a dominating component at 0.11 ns (78%) lead to a decreased average lifetime of 0.26 ns. These findings show that the photophysical properties of 1a/b are strongly influenced by the presence of the BHQ unit and that the properties of 1a$^+$ are unique within this series. This qualitatively reflects the behavior of the systems in $^1O_2$ production (1a$^+$ behaves differently than H-1a$^{2+}$, H-1b$^{2+}$, and 1b$^+$); however, there is no direct connection between the fluorescence lifetimes (decay from the singlet state) and the $^1O_2$ production (which occurs from the triplet state). Unfortunately, attempts to further characterize rotaxanes 1a/b, 2, and 3 by transient absorption spectroscopy were hampered by significant photobleaching of the ZnTPP upon irradiation. This issue resulted in a notable lack of reproducibility, so that no data regarding the triplet lifetimes could be obtained (see SI Table S1).

## Theoretical investigation of the conformational space of the rotaxanes

Although a pH-induced switching of singlet oxygen production for the H-1a$^{2+}$/1a$^+$ pair can be observed, it is reversed compared to what was initially envisaged in the design. Additionally, no switching is observed for the H-1b$^{2+}$/1b$^+$ pair even though NMR-spectroscopic analysis clearly shows that both systems do undergo a pH-induced relocation of the macrocycle. Thus, a better understanding of the molecular conformations of the rotaxanes 1a/b in both switching states was needed in order to explain their photophysical behavior. With the aid of computational chemistry, the conformational spaces of all four systems H-1a$^{2+}$/1a$^+$ and H-1b$^{2+}$/1b$^+$ were studied, focusing on possible intramolecular interactions involving the ZnTPP unit. In the first step, the potential energy surface of all variants was scanned using the fast semi-empirical methods GFN$n$-xTB ($n$ = 1, 2)[83–85] and the force field GFN-FF[86] in combination with the program CREST[87]. Subsequently, relevant (low-lying) and structurally representative conformations were selected and their Gibbs free energies (consisting of electronic gas-phase energy, thermal correction, and solvation contribution) were computed at a higher DFT level of theory (DFT: density functional theory)[88–90] (https://github.com/grimme-lab/xtb). For easier comparison, these conformers were divided into three categories: open, half-open, and closed. The division into those categories is based on binding to the Zn(II) center in its axial (octahedral) position and the folding of the whole system (see the methods sections or the SI for more details). Note that we could not consider explicit THF molecules in the calculations as this

would be computationally too demanding. The coordination of explicit THF molecules to the Zn(II) center is expected to be in competition with the $O_2$ molecules in the solution. However, the influence on the singlet oxygen production is most probably similar for both protonation states, so this effect cancels out for the presented comparison.

## Conformational search and Gibbs free energies for rotaxanes H-1a$^{2+}$/1a$^+$

In the investigation of the conformational spaces, almost all performed conformer searches for H-1a$^{2+}$ identified the same low-lying closed conformational motif indicating limited conformational flexibility. This is attributed to the high charge (+2) of the system, which leads to strong (intramolecular) electrostatic interactions that stabilize closed conformations. Finding the most relevant low-lying conformers of 1a$^+$ proved to be more challenging. A variety of distinct structural motifs were observed, with many of them featuring a different segment of the rotaxane coordinating with the Zn-atom. Moreover, several less folded (half-open) conformations comparable in energy to those of closed conformers were found, a phenomenon not observed for H-1a$^{2+}$. These results suggest a greater degree of flexibility for 1a$^+$.

Based on the results of the conformational searches, we selected relevant (low-lying) and structurally representative conformers and calculated the Gibbs free energies of a total of 31 conformers of H-1a$^{2+}$ and 26 conformers of 1a$^+$ (see SI Tables S4 and S5).

The computed electronic gas-phase, thermal, and solvation contributions varied strongly between closed, half-open, and open conformations, resulting in a considerable reranking of the investigated 1a$^+$ conformations (based on their Gibbs free energies), while the ranking stayed almost the same for H-1a$^{2+}$.

For H-1a$^{2+}$, primarily one closed structural motif was identified. Despite unfavorable solvation and thermal contributions, this conformation exhibited the lowest free energy (see Table S4 in the SI) and is depicted in Fig. 6a. As expected for this protonation state, the crown-ether macrocycle encircles the dibenzylammonium station, mediated by two NH···O hydrogen bonds (d(NH···O) = 1.84/1.95 Å) and three additional CH$_2$···O contacts (d(CH···O) = 2.45–2.86 Å). Tight molecular folding is enabled by a turn at the C$_2$H$_4$NMe linker between the triazolium group and the BHQ unit. Interestingly, this leads to a triple $\pi$-stack, involving the ZnTPP unit, the aniline group of the BHQ unit, and the triazolium group, which probably stabilizes this conformation. This unusual stacking, which places the triazolium group and the BHQ unit in close proximity to the ZnTPP, might also be responsible for the unusual NMR chemical shifts observed for the NMe group of the linker, the methoxy group of the BHQ moiety and the methyl group of the triazolium unit (vide supra). In general, most of the investigated closed conformations of this protonation state were thermodynamically preferred over (half-)open conformations (see SI Table S4). This result substantiates the previous finding that H-1a$^{2+}$ tends to adopt a strongly folded conformation in which the ZnTPP unit is coordinated intramolecularly.

However, for the deprotonated 1a$^+$ state, half-open conformations tend to be more stable than closed conformations (see Table S5 in the SI). Moreover, several distinct conformers were thermodynamically significant. This reaffirms the higher flexibility observed in the conformer searches. Two relevant conformers are depicted in Fig. 6b, c. These structures are energetically only separated by 8.7 kcal/mol, although their conformations are quite different: In both conformers, the crown-ether encircles the methyltriazolium station, as expected for the deprotonated rotaxane state. Here, the methyltriazolium group interacts with the macrocycle by $\pi$-stacking (for the lower-energy conformer only) and/or by CH···O interactions of the aromatic CH proton (both conformers, d(CH···O) = 2.30–2.82 Å). In the lower-energy conformer, all three aromatic units of the BHQ moiety showed $\pi$-contacts with other aromatic units. The terminal nitrophenyl group is located above the ZnTPP unit (as opposed to H-1a$^{2+}$, where the aniline group of BHQ is located above ZnTPP), whereas the dimethoxybenzene and the aniline moieties of the BHQ unit stack on top of the benzene rings of

**Fig. 6 | Side and top views of relevant conformers.**
**a** Conformers with the lowest Gibbs free energies of
**H-1a$^{2+}$**. **b** Conformers with the lowest Gibbs free
energies of **1a$^{+}$**. **c** Another relevant **1a$^{+}$** conformer
with its free energy relative to the lowest con-
formation. The coloring of the carbon atoms is
equivalent to Fig. 2. For better clarity, the triazolium
station is not colored, and all H-atoms (except for
NH$_2$) are excluded.

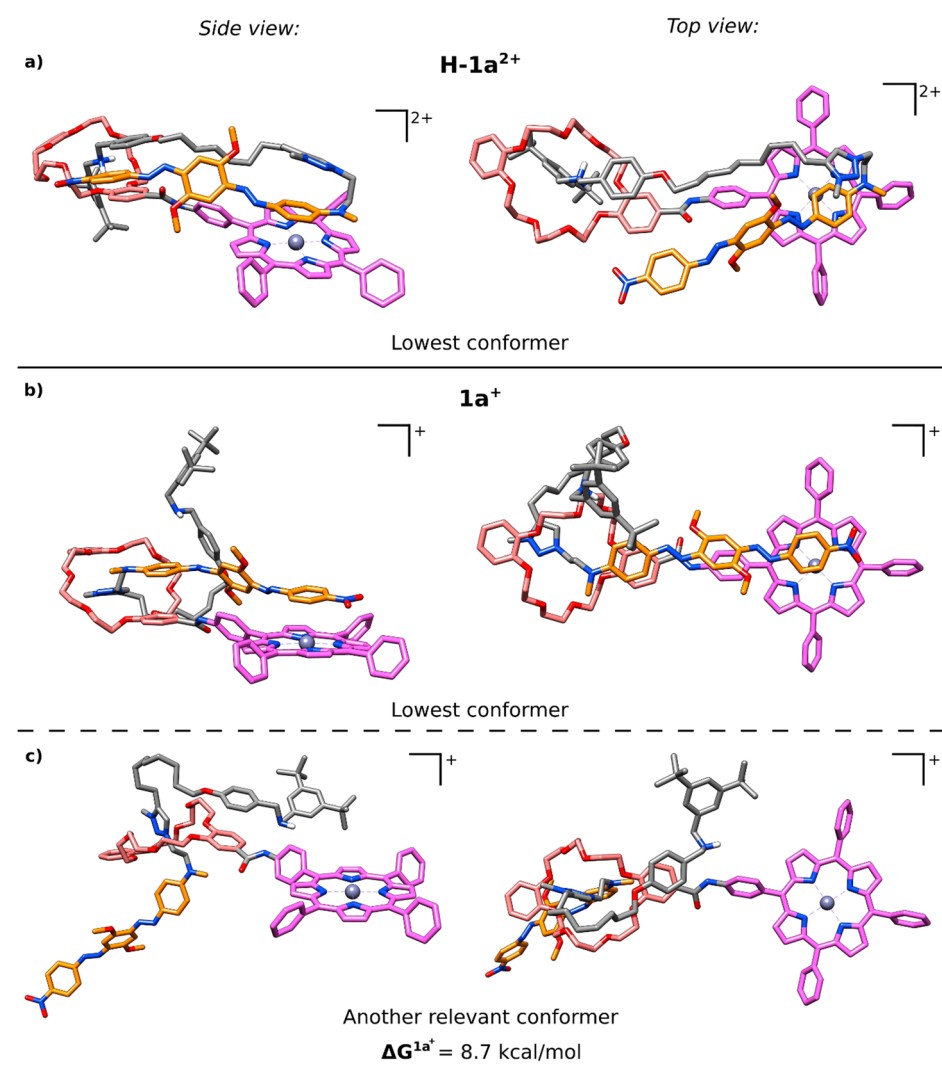

ZnTPP and the dibenzo[24]crown[8] macrocycle, respectively. In contrast,
the higher-energy conformer shows no intermolecular interactions invol-
ving the ZnTPP group, and the entire BHQ unit is pointed away from the
ZnTPP (see Fig. 6c).

In conclusion, **H-1a$^{2+}$** is mostly found in one dominant, strongly fol-
ded (*closed*) conformation, with the ZnTPP unit being coordinated by the
BHQ unit. In contrast, **1a$^{+}$** also adopts conformers that are in a *half-open*
conformation and proved to be more flexible. As a consequence, the ZnTPP
unit in **1a$^{+}$** is less frequently in close spatial proximity to the BHQ moiety.
This may influence the $^1O_2$ production of the rotaxane in two ways: First, a
close proximity of ZnTPP and the quencher will facilitate energy transfer
from the ZnTPP triplet excited state, thus inhibiting $^1O_2$ production. Sec-
ond, a direct contact of the BHQ with the ZnTPP might also inhibit dif-
fusion of $^3O_2$ towards ZnTPP, thus preventing energy transfer to $^3O_2$. The
computational data is therefore consistent with experimental findings: The
strongly folded conformation of **H-1a$^{2+}$** leads to an almost complete shut-
down of the $^1O_2$ production ($\Phi_\Delta$= 0.03, c.f. $\Phi_\Delta$ = 0.72 for free ZnTPP, see
Table 1). In **1a$^{+}$** there are multiple accessible conformations, only some of
which feature a close contact between ZnTPP and the BHQ unit. This leads
to a considerable, but not complete quenching of the $^1O_2$ production
($\Phi_\Delta$ = 0.15) for this protonation state. Together this results in the reversed
direction of switching compared to the original design concept.

To further substantiate these qualitative correlations, we also briefly
investigated whether the computed structures of **H-1a$^{2+}$** and **1a$^{+}$** lead to a
Förster radius comparable to what is experimentally expected. Due to the
size of the BHQ unit, which features three aromatic rings connected by two

diazo groups, it was unclear which position of the BHQ unit should be used
to measure intramolecular distances. Thus, this attempt was discarded (for
more details, see SI section 2.3).

## Conformational search and Gibbs free energies for rotaxanes H-1b$^{2+}$/1b$^{+}$

We also examined the short variants **H-1b$^{2+}$** and **1b$^{+}$** using the same
approach described above. The conducted simulations showed a reduced
flexibility for both variants, which likely resulted from the shorter alkylene
spacer. Mostly *closed* conformations were found, with many binding motifs
being similar for both protonation states. Gibbs free energies of 16 con-
formers of **H-1b$^{2+}$** and 17 conformers of **1b$^{+}$** were calculated (see SI Table S6
and Table S7). We found that *closed* conformations are thermodynamically
favorable for both **H-1b$^{2+}$** and **1b$^{+}$**, and their Gibbs free energy distributions
are considerably more similar than those of **H-1a$^{2+}$** and **1a$^{+}$**. Thus, for both
**1b** protonation states, the ZnTPP is in close contact with the BHQ unit,
suggesting a low singlet oxygen quantum yield. This result is in line with
experimental findings, where **H-1b$^{2+}$** and **1b$^{+}$** showed the same vanishing
$^1O_2$ production ($\Phi_\Delta$ < 0.01/0.01), similar to that of **H-1a$^{2+}$**.

## Conclusions

Based on our original concept for a pH-switchable rotaxane for singlet
oxygen production, we have conducted an in-depth study that highlights the
importance of molecular folding in the design of rotaxane-based switches.

We synthesized rotaxanes **1a/b**, which consist of a ZnTPP-appended
macrocycle and a thread featuring a BHQ unit as one of the stoppers.

Both systems underwent pH-induced switching leading to reversible relocation of the macrocycle between an amine/ammonium and a triazolium station, as evidenced by NMR. For rotaxane **1b**, with a shorter $C_3$ linker, no $^1O_2$ production was observed independent of the switching state. However, for rotaxane **1a**, which features a longer $C_9$ linker between both stations, the switching has a strong effect on $^1O_2$ production, with the deprotonated rotaxane **1a$^+$** representing the *on*-state ($\Phi_\Delta = 0.15$), while the protonated rotaxane **H-1a$^{2+}$** represents the *off*-state ($\Phi_\Delta = 0.03$). Thus, we achieved the important goal of generating a system that can be switched *off* almost completely, minimizing unwanted side-effects of $^1O_2$, while the *on*-state is significantly more active (500% increase $^1O_2$ production). However, the direction of switching is inverted with regard to the original design, where we placed the BHQ unit close to the triazolium station, assuming that in the deprotonated rotaxane **1a$^+$** the positioning of the macrocycle around the triazolium would lead to suppressed $^1O_2$ production (and vice versa).

This unexpected behavior (no $^1O_2$ production for either **1b$^+$/H-1b$^{2+}$**, $^1O_2$ production for **1a$^+$** but not for **H-1a$^{2+}$**) was explained by in-depth theoretical calculations of the conformational space of these systems: In the protonated state **H-1a$^{2+}$**, the ZnTPP unit does not have a larger distance from the BHQ-stopper (as intended), but intramolecular folding actually leads to one energetically preferred closed conformation with a strong interaction between the ZnTPP unit and the BHQ moiety. In contrast, **1a$^+$** adopts several energetically similar conformations, including half-open conformations without intramolecular contacts with ZnTPP. This explains why **1a$^+$** represents the *on*-state of the system, based on the ability of the ZnTPP to mediate $^1O_2$ production without interference from the quencher. As for **1b$^+$/H-1b$^{2+}$**, the conformational analysis also showed mostly closed conformations with intramolecular interactions between ZnTPP and BHQ, thus explaining suppressed $^1O_2$ photosensitization for these rotaxanes.

These studies highlight the importance of taking into account molecular flexibility when designing functional switchable systems, such as switchable photosensitizers (as in this study), as well as other applications. Especially for highly charged species (such as **H-1a$^{2+}$/H-1b$^{2+}$**) that can additionally interact via favorable intramolecular interactions such as π-stacking, strongly folded conformations can be preferred. In the present case, even seemingly small structural units, such as the $C_2H_4NMe$ linker between the triazole and the BHQ unit, can have a significant impact if they allow the formation of energetically relevant folded structures.

Thus, we have not only presented the synthesis of pH-switchable rotaxanes for singlet oxygen production, but also developed a deeper understanding of the design of bistable rotaxanes for the development of functional switchable systems in general.

## Methods
### Materials
For a list of the commercially available chemicals used, see SI section 1.1.1. Known compounds were prepared according to literature procedures (see SI section 1.2.1, Figs. S1–S3).

### Standard analytical methods
NMR spectra were recorded with a Bruker Avance NEO 400 or a Bruker DRX 600 spectrometer at 298 K using CDCl$_3$, CD$_2$Cl$_2$, THF-$d_8$, or DMSO-$d_6$ as the solvent. The chemical shifts are referenced relative to the residual proton signals of the solvent in $^1$H-NMR or the signal of the solvent in $^{13}$C-NMR. Further instrumental details are given in section 1.1.2 of the SI. Full characterization details of all new compounds are given in section 1.2.2 of the SI. For NMR spectra of all new compounds, see section 3 of the SI (Figs. S49–S68). UV/Vis-absorption spectra were recorded on a Varian Cary 300 Bio UV-Vis spectrophotometer in spectrophotometric grade tetrahydrofuran. Fluorescence spectra were recorded on a Varian Eclipse fluorescence spectrophotometer. Time-resolved measurements were carried out on a FluoTime 300 spectrometer from PicoQuant.

### Synthesis of rotaxanes
All rotaxanes (**1a/b, 2** and **3**) were prepared by the following general procedure: Dibenzylammonium half-thread **4a/b-HPF$_6$** (1 eq.), crown-ether macrocycle **5** (1.2 eq.) and the corresponding azide (1.3 eq.) were dissolved in DCM (4 mL per mmol of **4a/b-HPF$_6$**), degassed by purging with argon for five minutes and stirred with tetrakis(acetonitrile)copper(I) hexafluorophosphate (1.5 eq.) overnight. The rotaxanes were purified by flash column chromatography (SiO$_2$, DCM:MeOH). For methylation of the triazole, the rotaxanes were stirred in methyl iodide (0.1 mL/mg) until full conversion of the starting material was observed by thin layer chromatography (2-10 days). Excess methyl iodide was removed in vacuo and the methylated compound was purified by flash column chromatography (SiO$_2$, DCM:MeOH) if necessary. The methylated rotaxane was then dissolved in DCM (2 mL) and stirred over solid ammonium-hexafluorophosphate (20 eq.) for 20 hours, filtered over a polyamide syringe filter, and dried under reduced pressure. For details, see section 1.2.2 of the SI.

### Switching of rotaxanes
The protonated rotaxane-bishexafluorophosphate was dissolved in acetone (2 mL/10 mg), and 50 mM aqueous sodium hydroxide (2 eq.) was added. Volatiles were removed *in vacuo* and the residue was taken up in dichloromethane and filtered over a polyamide syringe filter. The filtrate was evaporated to yield the deprotonated rotaxane (see SI Figs. S12–S18).

### Investigation of singlet dioxygen photoproduction
Approximately 10 μM solutions of free porphyrin or the rotaxanes were prepared in THF under ambient conditions and the absorbance of the most intense band (*Soret* band around 424 nm) was adjusted to 0.8. Then, 70 μL DPBF solution (860 μM) was added in the dark. Then the cuvette was irradiated with 420 nm light in intervals of 10 or 30 seconds and fluorescence emission spectra ($\lambda_{Exc} = 420$ nm) were recorded to monitor the decrease in fluorescence intensity ($I$) of DPBF at 457 nm over time (see SI Fig. S19).

### Photophysical characterization
Steady-state excitation and emission spectra were recorded on a FluoTime 300 spectrometer from PicoQuant (a full description of the equipment can be found in section 1.4.3 of the SI). Steady-state spectra and photoluminescence lifetimes were recorded in TCSPC mode by a PicoHarp 300 (minimum base resolution 4 ps). Emission and excitation spectra were corrected for source intensity (lamp and grating) by standard correction curves. An instrument response function calibration was performed using a diluted Ludox® dispersion. Lifetime analysis was performed using the commercial EasyTau 2 software (PicoQuant). The quality of the fit was assessed by minimizing the reduced chi-squared function ($\chi^2$) and visual inspection of the weighted residuals and their autocorrelation (for further details, see SI section 1.4.3, Figs. S25–S42).

### Computational details
We employed the *Conform-Rotamer Ensemble Sampling Tool* CREST[87] (v. 2.12) in combination with the semi-empirical quantum mechanical (SQM) methods GFN$n$-xTB ($n$ = 1, 2)[83–85] and the force field GFN-FF[86], applying the implicit solvation model ALPB (THF)[91]. Chosen structures were optimized at the GFN2-xTB [ALPB:THF] level of theory. Thermal corrections ($G_{thermo}$) were computed at the same level by employing the modified rigid-rotor-harmonic-oscillator approximation (mRRHO)[88]. This was done using the *xtb* (v. 6.5.1) program (https://github.com/grimme-lab/xtb). Utilizing the *TURBOMOLE* (v. 7.5.1) (https://www.turbomole.org)[92] program package, electronic gas-phase energies ($E_{gas}$), and solvation contributions ($\partial G_{solv}$) were computed using the PBEh-3c composite DFT method[89]. The solvation contributions were computed

with the COSMO-RS[90] implicit solvation model. Finally, Gibbs free energies ($G_{Gibbs}$) were calculated using:

$$G_{Gibbs} = E_{gas} + G_{thermo} + \partial G_{solv}$$

To facilitate comparison, only relative values ($\triangle X = X - X_{ref}$) are discussed, with the smallest value being defined as the respective reference value ($X_{ref}$).

Division of the found conformations into three categories *open*, *half-open*, and *closed* is based on the degree of intramolecular folding exhibited by the structures. As a quantification the computed *S*olvent *A*ccessible *S*urface *A*rea (SASA) of each conformer was used (https://github.com/grimme-lab/numsa). In the *open* conformers no or minimal folding is observed, and the Zn atom in the ZnTPP unit is not coordinated in its axial (octahedral) plane. The *closed* conformations exhibit a high degree of intramolecular folding, and (mostly) a coordination of the Zn atom in the aforementioned plane. *Half-open* conformers show a moderate degree of folding, but in some cases still exhibit a coordinating group on the Zn atom. For more details, see section 2.1 of the SI.

## Data availability

The Supplementary information contains all details of the synthesis and characterization of novel compounds, photophysical measurements, determination of singlet oxygen production, and details regarding the theoretical calculations and their evaluations. Supplementary data 1 include the xyz-files of the calculated conformers of the rotaxanes. Supplementary data 2 include the numerical source data for the determination of the singlet oxygen quantum yields.

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

## Acknowledgements
J.N. would like to thank the Deutsche Forschungsgemeinschaft DFG (project NI1273/2-2 and Heisenberg-Professorship NI1273/4-1) for funding. C.A.S. gratefully acknowledges funding from the Deutsche Forschungsgemeinschaft (DFG, German Research Foundation)—project-ID 433682494 - SFB 1459 "Intelligent Matter". C.A.S. gratefully acknowledges the generous financial support for the acquisition of an "Integrated Confocal Luminescence Spectrometer with Spatiotemporal Resolution and Multiphoton Excitation" (DFG/Land NRW: INST 211/915-1 FUGG; DFG EXC1003: Berufungsmittel) and of a "Laser-Induced Transient Absorption and Photoacoustic Spectrometer" (DFG/Land NRW: INST 211/1033-1 FUGG; DFG EXC1003: Berufungsmittel).

## Author contributions
J.R. performed the synthetic chemistry. J.R., M.E.G.S., and M.B.R.A. performed the photophysical characterization and singlet dioxygen measurements. B.B. and S.L. performed the theoretical calculations. J.R. and J.N. devised the project. C.A.S., S.G., and J.N. directed the research. All authors contributed to the writing of the manuscript.

## Funding

## Competing interests
The authors declare no competing interests.
