## [Peer Review File · Communications Chemistry]

Reviewers' comments:

Reviewer #1 (Remarks to the Author):

The authors report a new molecular interlocked system that can be used to control the singlet oxygen production of a fluorophore by regulating its quenching via pH. The authors combine synthesis spectroscopy and computational modelling to explain the counterintuitive results they observed when testing the 1O_2 production in their system upon pH variation.

Overall the presentation, execution and interpretation of the work is outstanding, the results are novel and could potentially affect the MIMs community. This work should be accepted in CommsChem after revision of a few, very minor, points.

In Figure 6 the authors show that Conformer 2 (interpretable as the conformer with the second lower energy form the ones analyzed) is 8.7 kcal/mol in energy higher than the lowest one. However, following this interpretation and looking at Table S5, entry 13 is a half-open conformer with 1.17 kcal/mol of relative ΔG , way lower in energy than Conformer 2 in the picture. Moreover, if Conformer 2 would mean entry number 2 in Table S5 (ordered according to SASA) then also this value would be wrong (13.21 Kcal/mol). There is an entry 16 with 8.72 kcal/mol, however, the authors should double check this point and in case explain why did they choose this entry at all when there are two half open structures and an open structure higher in energy than the one presented.

The authors should explain how the “representative structures” (page S46) were chosen (e.g. randomly, using PCA clustering, following the energy of the CREST conformer output...). The authors show the vast range of options that they have used to explore the conformational analysis of the rotaxanes (Table S2), but it is not immediately clear from which run (if any) the starting conformers that were then reoptimized were taken.

Still on Table S2, while it is possible to understand what the authors mean with simulation length, maybe they could specify that this is the MTD length.

I would like the authors to include an additional comment on the Förster radius estimation. The experimental value is estimated by implying a free rotation of the dye and the quencher with $k_2=2/3$. However, this is not the case for the system under analysis, where the dye and the quencher are closely linked. Hence, the value of 2.8 Å could be the subject of debate. I understand that estimating a more correct value for k_2 is quite a complex task by itself and often relies on computational estimations, making section 2.3 of the SI - where the authors try to estimate the

Förster radius computationally comparing it to the experimental value based on $k_2=2/3$ - a dog that bites its own tail. Hence comparing the two values is by itself not completely correct.

In addition to the previous comments, the authors analyze the distances between the quencher and the fluorophore to estimate the Förster radius. I would like to ask here two curiosities (and as such, these curiosities do not necessitate taking action if the time to test them is deemed to be unreasonable). Would it be possible to use the overlaps of the surface generated by the van der Waals radii of the atoms (while being as approximate as the distance compared to the orbital overlaps, it could provide a more balanced description of the overlap itself than the distances at selected fixed points) as a way to put a weight on the computationally estimated Förster radii? In this way, the authors could maybe take the distance between the center of mass of the dye and the quencher and apply the weight of the vdW overlaps to estimate the radius.

More importantly for the paper: are the means in Tables S8 and S9 Boltzmann averaged? And if so, does applying the average make a difference?

Reviewer #2 (Remarks to the Author):

In the manuscript submitted by Riebe et. al. switchable singlet oxygen production by a series of rotaxanes are examined. A ZnTPP photosensitizer is attached to macrocycle and one of the stopper is chosen to be a quencher. Acid-base dependent relocation of the macrocycle between ammonium and triazolium stations are shown to change the conformation of the molecule which depends on the flexibility provided by linker size. Although singlet oxygen switching with interlocked systems are studied previously, this work may contribute to the conformational dynamics of this process with a new molecular structure. Following issues should be addressed by authors:

1. Lifetime analysis with deprotonated rotaxane gives multiple lifetimes one of which is very close to the lifetime of free ZnTPP. Authors should discuss the possibility that macrocycle can slip over BHQ stopper, resulting in the generation of free macrocycle or a perched complex with the thread. If this is the case, then increase in singlet oxygen quantum yield in deprotonated state may be due to this free ZnTPP. In order to test this authors are suggested to check the barrier size with unmethylated rotaxane. Unmethylated rotaxane can be deprotonated and formation of macrocycle can be checked by mass spectrometry analysis or NMR. H-1a might be a pseudorotaxane kinetically trapped and deprotonation may lead to dethreading.

2. Thread of compound 1 should also be synthesized (if possible) and NMR spectra should be compared in protonated and deprotonated states. Thread would also give information about conformational dynamics and stability of the molecule.
3. NMR spectra of protonated and deprotonated rotaxanes are taken in methylene chloride (after acid-base treatment in acetone), however singlet oxygen production experiments are done in THF. Solvent would interfere with the conformational dynamics and switching behaviour. Therefore, singlet oxygen experiments should also be done for compound 1a, H1a in dichloromethane.
4. Singlet oxygen production is analysed with pure compounds. This switching behaviour should also be tested in situ by adding acid or base and checking 1O_2 production without any prior purification. A reference DPBF solution should also be tested to eliminate acid-base dependent change in DPBF response.
5. In Figure 3, after deprotonation H3 and H4 peaks are still observed around 4.6 ppm. Does this mean incomplete deprotonation? Authors should explain this.

Reviewer #3 (Remarks to the Author):

In this nice manuscript, bistable rotaxanes were used to develop pH-switchable systems for singlet dioxygen photoproduction. Based on the combination of a Zn(II)-tetraphenylporphyrin photosensitizer, which was attached to the macrocycle, and a black-hole-quencher, which was used as one of the rotaxane-stoppers, singlet oxygen production could be switched on/off ($\Phi\Delta = 0.15/0.03$) by the addition of base/acid. The authors found that only a sufficiently long linker between both stations on the thread enabled switchability, and that the direction of switching was inversed with regard to the original design. This unexpected behavior was attributed to intramolecular folding of the rotaxanes, as indicated by extensive theoretical calculations. This evidences the importance to take into account the conformational flexibility of large molecular structures when designing functional switchable systems. This manuscript, with good novelty and scientific value, can be published as it is.

Answers to the reviewer comments:**Reviewer 1:**

The authors report a new molecular interlocked system that can be used to control the singlet oxygen production of a fluorophore by regulating its quenching via pH. The authors combine synthesis spectroscopy and computational modelling to explain the counterintuitive results they observed when testing the $^{1}O_2$ production in their system upon pH variation.

Overall the presentation, execution and interpretation of the work is outstanding, the results are novel and could potentially affect the MIMs community. This work should be accepted in CommsChem after revision of a few, very minor, points.

We thank the reviewer for this very positive evaluation of our work.

Question 1:

In Figure 6 the authors show that Conformer 2 (interpretable as the conformer with the second lower energy form the ones analyzed) is 8.7 kcal/mol in energy higher than the lowest one. However, following this interpretation and looking at Table S5, entry 13 is a half-open conformer with 1.17 kcal/mol of relative ΔG , way lower in energy than Conformer 2 in the picture. Moreover, if Conformer 2 would mean entry number 2 in Table S5 (ordered according to SASA) then also this value would be wrong (13.21 Kcal/mol). There is an entry 16 with 8.72 kcal/mol, however, the authors should double check this point and in case explain why did they choose this entry at all when there are two half open structures and an open structure higher in energy than the one presented.

Answer:

We thank the reviewer for the helpful feedback. We agree that the chosen way of naming the conformers in the main paper led to confusion. As correctly observed the “conformer 2” in the main paper corresponds to conformer 16 in the SI and is not the second lowest conformer found. (Reminder: the conformers in the SI are ordered by their SASA and not their energies). To avoid confusion, we adjusted the text in figure 6 to “Another relevant conformer” and added information in Table S5 that conformers 16 and 9 are the ones shown in the main paper.

Concerning our decision to show this conformation instead of one lower in energy (see Table S5): To avoid making the manuscript unnecessarily long, we decided to only show two conformers for $1a^+$ (and only one for $H-1a^{2+}$), which led to the problem of picking representative conformers. Conformers 13 and 14 are similar to the lowest conformer (i.e. same coordinating group to Zn) which is already shown (Figure 6b). Open conformers 22 and 26 are similar to each other and can be thought of as 3D structures of the 2D representations shown in Figure 2. Considering the error of the used methods, conformers 22, 26 and 16 essentially have the same Gibbs free energy (< 1 kcal/mol difference). We preferred to show the latter as we deemed it chemically more interesting, and because all other structure motifs are already included in the manuscript. We hope that the reviewer can agree with this line of thought.

Question 2:

The authors should explain how the “representative structures” (page S46) were chosen (e.g. randomly, using PCA clustering, following the energy of the CREST conformer output...).

Answer:

We thank the referee for the remark. To make this clearer for the reader we added the following remark in the SI of the manuscript on page S50:

“To investigate potential shortcomings of the SQM methods and the low-level implicit solvation model, sets of relevant and representative structures were created for both protonation states, and investigated further at a higher level of theory. Low-lying conformations of the different CREST simulations were added to these sets, unless a similar structural motif was already included, or their energies were too high compared to the other conformations in the set. However, due to the large system size, these low-lying conformations in the ensembles created by CREST often included multiple different variations of the same structural motifs, therefore we also investigated molecular dynamic and meta-dynamic simulations performed during the CRE generations and manually picked out additional conformations that seemed of interest. Such conformations especially featured distinct binding motifs of the Zn atom by the quencher. Since the representative structures were evaluated on a DFT level, our goal was to prevent the neglect of a binding motif due to an underestimation of the previously used SQM methods. Additionally, conformations with almost no intramolecular folding were taken randomly out of different meta-dynamic simulation runs performed by CREST as such motifs are expected to be underestimated by the cheap solvation model used in the sampling.”

Question 3:

The authors show the vast range of options that they have used to explore the conformational analysis of the rotaxanes (Table S2), but it is not immediately clear from which run (if any) the starting conformers that were then reoptimized were taken.

Answer:

For the discussed system size and its flexibility, the conducted simulations with CREST are not deterministic. CREST calculations from the same start structure may (or may not) lead to different conformer/rotamer ensembles, which holds true for **1a⁺** (and not for **H-1a²⁺**). Therefore, naming the exact origin of the starting structures will not give any relevant information. Choosing the starting conformers for further reoptimization is made clearer in the SI, thanks to the former question (see page S50).

Question 4:

Still on Table S2, while it is possible to understand what the authors mean with simulation length, maybe they could specify that this is the MTD length.

Answer:

We thank the reviewer for his suggestion and agree that this information leads to more clarity. We added this in the SI, Table S2.

Question 5:

I would like the authors to include an additional comment on the Förster radius estimation. The experimental value is estimated by implying a free rotation of the dye and the quencher with $k_2=2/3$. However, this is not the case for the system under analysis, where the dye and the quencher are closely linked. Hence, the value of 2.8 Å could be the subject of debate. I understand that estimating a more correct value for k_2 is quite a complex task by itself and often relies on computational estimations, making section 2.3 of the SI - where the authors try to estimate the Förster radius computationally comparing it to the experimental value based on $k_2=2/3$ - a dog that bites its own tail. Hence comparing the two values is by itself not completely correct.

Answer:

While the reviewer is correct that taking $\kappa^2 = 2/3$ does not result in a "correct" value for R_0 , we would argue that inclusion of this value serves to illustrate the point that a sufficiently long linker should have led to a strong differentiation between the switching states, which is why it was the guide for our original design. Furthermore, other researchers have shown that using $\kappa^2=2/3$ still leads to a value for R_0 with an acceptable error of $\pm 15-20\%$ even for somewhat rigid dyads. (see for example: <https://doi.org/10.1021/jp0111968> and <https://doi.org/10.1021/ja105725e>).

For example, assuming $\kappa^2 = 0.5$ results in $R_0 = 2.7$ nm and for $\kappa^2 = 1$ we find $R_0 = 3.0$ nm, both well within $\pm 20\%$ of the original estimate. In absence of a better value for κ^2 (the estimation of which, we agree, would be outside the scope of this work) we believe that the value of $R_0 = 2.8$ nm still serves as a somewhat useful guide. The attempt to compare this value with the distances found in the simulated structures is therefore still a valid approach. We have added the mentioned references, as well as the above-mentioned errors of the estimation in section 1.4.2 of the SI.

Question 6:

In addition to the previous comments, the authors analyze the distances between the quencher and the fluorophore to estimate the Förster radius. I would like to ask here two curiosities (and as such, these curiosities do not necessitate taking action if the time to test them is deemed to be unreasonable). Would it be possible to use the overlaps of the surface generated by the van der Waals radii of the atoms (while being as approximate as the distance compared to the orbital overlaps, it could provide a more balanced description of the overlap itself than the distances at selected fixed points) as a way to put a weight on the computationally estimated Förster radii? In this way, the authors could maybe take the distance between the center of mass of the dye and the quencher and apply the weight of the vdW overlaps to estimate the radius.

Answer:

Thank you very much for the interesting idea. We have tested this approach and visually observed a considerable overlap of the "vdW spheres" for closed conformations. As a fast attempt to quantify this, we used the solvent-accessible surface areas (SASA) for specific cutouts. In these cutouts we only included the ZnTPP unit (cut after the benzene of the porphyrin unit) and the BHQ unit (cut at the nitrogen defined as the starting point of BHQ for the Förster radii). Then we computed the SASA of each species of the cutout and the SASA of the whole cutout (hence including the ZnTPP and BHQ units). If the vdW spheres show a large overlap between

the two units then the SASA of the whole cutout is smaller than the sum of the SASAs of the two units. By this approximate approach, we were able to obtain a quantitative description of the surface overlap of the vdW spheres. The overlap of many open conformations is 0, and/or very small for some of the half-open conformers. Therefore, using this as a scaling factor mostly puts weight on the closed conformation, which, in our opinion, distorts the picture.

Besides these points, the issues described in the supporting information persist (missing explicit solvation, spatial extension of the quencher). We therefore believe that further investigations regarding this topic are out of the scope of this study.

Question 7:

More importantly for the paper: are the means in Tables S8 and S9 Boltzmann averaged? And if so, does applying the average make a difference?

Answer:

We thank the reviewer for the remark. The means in Tables S8 and S9 are not Boltzmann averaged. We have added this information to the table captions.

Using Boltzmann factors (with $T=293.15\text{K}$), the contributions of all conformers except the lowest ones found are negligible. Boltzmann averaging works best for an (almost) complete conformer/rotamer ensemble, which is not present here. Furthermore, the rather large size of the systems ruled out the application of highly accurate methods (e.g. hybrid DFT in a large basis), so the possible error in the Gibbs energies might have a significant impact on the Boltzmann weighting.

Reviewer 2:

In the manuscript submitted by Riebe et. al. switchable singlet oxygen production by a series of rotaxanes are examined. A ZnTPP photosensitizer is attached to macrocycle and one of the stopper is chosen to be a quencher. Acid-base dependent relocation of the macrocycle between ammonium and triazolium stations are shown to change the conformation of the molecule which depends on the flexibility provided by linker size. Although singlet oxygen switching with interlocked systems are studied previously, this work may contribute to the conformational dynamics of this process with a new molecular structure.

We thank the reviewer for this very positive evaluation of our work.

Question 1:

Lifetime analysis with deprotonated rotaxane gives multiple lifetimes one of which is very close to the lifetime of free ZnTPP. Authors should discuss the possibility that macrocycle can slip over BHQ stopper, resulting in the generation of free macrocycle or a perched complex with the thread. If this is the case, then increase in singlet oxygen quantum yield in deprotonated state may be due to this free ZnTPP. In order to test this, authors are suggested to check the barrier size with unmethylated rotaxane. Unmethylated rotaxane can be deprotonated and formation of macrocycle can be checked by mass spectrometry analysis or NMR. H-1a might be a pseudorotaxane kinetically trapped and deprotonation may lead to dethreading.

Answer:

We thank the reviewer for this insightful comment. While it is true that in principle all rotaxanes can be considered kinetically trapped pseudorotaxanes that dissociate with sufficient input of energy, we believe that this is not a case of concern for the rotaxanes in this study for the following reasons: Firstly, in some instances the rotaxanes were additionally analyzed in DMSO- d_6 prior to methylation and re-isolated by rotary evaporation at 70 °C without any sign of decomposition into non-interlocked components. Secondly, the mass spectra of sufficiently pure rotaxanes before or after methylation show no signals corresponding to the non-interlocked components, which would be expected if they could easily dissociate. Finally, the reversible switching shown in Fig. 3 of the main article rules out dethreading of a pseudo-rotaxane since it is known that the triazolium can only act as a binding station in an interlocked rotaxane-structure. Thus, in a kinetically trapped pseudorotaxane, deprotonation would lead to dethreading. In such case, re-threading after reprotonation would also be kinetically hindered and signals of the free thread and free macrocycle in addition to the rotaxane would all be observed as separate species, similar to the true pseudorotaxane in Fig. S17 in the SI. Because this is not observed, we are sure that the investigated rotaxanes are stable under our experimental conditions.

Question 2:

Thread of compound 1 should also be synthesized (if possible) and NMR spectra should be compared in protonated and deprotonated states. Thread would also give information about conformational dynamics and stability of the molecule.

Answer:

Unfortunately, repeated attempts to prepare the methylated thread of rotaxane **H-1a**²⁺ failed. From the reaction mixture of the rotaxanation, some of the unmethylated thread could be isolated. For making the methylated thread, the dibenzylamine-group was Boc-protected to allow regioselective methylation of the triazole. Unfortunately, following the protocol for methylation of the triazole established for the rotaxanes, only a fragment of the desired thread containing the BHQ-2-chromophore, the triazolium-station and the long alkyl-chain could be isolated, indicated by NMR and mass spectrometry. We suspect that trace amounts of hydroiodic acid present in the methyl iodide caused cleavage of the Boc-group which allowed for permethylation of the dibenzylamine, transforming it into a leaving group which led to decomposition at the dibenzylamine-position. The fact that such decomposition is not observed for methylation of the rotaxanes serves as a further testimony to their stability and the ability of the crown-ether macrocycle to prevent methylation of the ammonium-station. We have added a sentence about the unsuccessful synthesis of the thread in the main article (page 6).

However, since the rotaxanes are stable (see above reply) and switching in THF can also clearly be shown (see below), even without comparison to the spectrum of the thread, we believe the evidence for our conclusions to be sufficient.

Question 3:

NMR spectra of protonated and deprotonated rotaxanes are taken in methylene chloride (after acid-base treatment in acetone), however singlet oxygen production experiments are done in THF. Solvent would interfere with the conformational dynamics and switching behaviour. Therefore, singlet oxygen experiments should also be done for compound 1a, H1a in dichloromethane.

Answer:

We thank the reviewer for this valuable comment, comparing photochemical results and NMR-spectra in the same solvent would further strengthen the conclusions we draw from our investigations. However, instead of performing the photochemical experiments in another solvent which would also necessitate new photophysical characterization, we investigated the switching behavior of rotaxane **H-1a**²⁺ by NMR in THF-d₈. Pleasingly, the characteristic shifts discussed in the main article can clearly be reproduced, unambiguously proving that translocation of the macrocycle occurs also in THF solution with the same efficiency as in DCM. Additionally, also for the rotaxane S20 containing an unsubstituted DB24C8-macrocycle and an anthracene-stopper, switching in THF-d₈ solution can clearly be shown by NMR. We have added spectra for both compounds to the SI (figures S13 and S18). We hope that these findings, in combination with the report by Credi cited in the reply to the Editor (see below), can fully address the concerns brought forth in this comment.

Question 4:

Singlet oxygen production is analysed with pure compounds. This switching behaviour should also be tested *in situ* by adding acid or base and checking ¹O₂ production without any prior purification. A reference DPBF solution should also be tested to eliminate acid-base dependent change in DPBF response.

Answer:

While we value the idea expressed in this comment, we think that adding yet more variables to the experiment will only unnecessarily increase the complexity of the system. We have attempted to use soluble amine bases like pyridine and triethylamine for *in-situ* switching, but have found that they themselves act as quenchers for singlet oxygen production, presumably by coordination to the Zn-center in the porphyrin. Therefore, we purposefully performed the deprotonation *ex-situ*, removing excess base and formed salts to enable observation of the effects of the switching in an isolated fashion and not having to control for interactions with other species in solution. The fact that switching did not significantly influence singlet oxygen production for rotaxanes **2** and **3** shows that basicity/acidity of the rotaxane-structure itself does not play a role in the observed behavior in DPBF-consumption for rotaxane **1a**, making further control experiments involving DPBF plus acid/base unnecessary in our opinion.

Additionally, we would like to note that spectrum c) in Fig. 3 of the main article was obtained by protonation *in-situ*, showing that both methods lead to the same switching state.

Question 5:

In Figure 3, after deprotonation H3 and H4 peaks are still observed around 4.6 ppm. Does this mean incomplete deprotonation? Authors should explain this.

Answer:

As indicated in Fig. 3 in the main article, the signals of H3 and H4 after deprotonation can clearly be assigned and are shifted significantly upfield. With regard to the signals observed at around 4.5 ppm, we currently believe that these signals belong to an aromatic group of the quencher, featuring a strong upfield shift due to stacking with the ZnTPP. This would be in line with the fact that such signals cannot be observed for the deprotonated versions of the control rotaxanes **H-2⁺**, **H-3⁺** and **S20⁺**, which don't feature the BHQ-unit (see SI figures S15/16/17). Unfortunately, we could not clearly assign these signals due to missing crosspeaks in the 2D-NMR data, but we are certain that these signals are not due to incomplete deprotonation. We have added a sentence to emphasize this in the description of the figure 3 of the main paper.

Reviewer 3:

In this nice manuscript, bistable rotaxanes were used to develop pH-switchable systems for singlet dioxygen photoproduction. Based on the combination of a Zn(II)-tetraphenylporphyrin photosensitizer, which was attached to the macrocycle, and a black-hole-quencher, which was used as one of the rotaxane-stoppers, singlet oxygen production could be switched on/off ($\Phi\Delta = 0.15/0.03$) by the addition of base/acid. The authors found that only a sufficiently long linker between both stations on the thread enabled switchability, and that the direction of switching was inversed with regard to the original design. This unexpected behavior was attributed to intramolecular folding of the rotaxanes, as indicated by extensive theoretical calculations. This evidences the importance to take into account the conformational flexibility of large molecular structures when designing functional switchable systems. This manuscript, with good novelty and scientific value, can be published as it is.

We thank the reviewer for this positive evaluation of our work and hope that the other reviewers will agree with this comment after we have improved the manuscript by performing the aforementioned changes.

REVIEWERS' COMMENTS:

Reviewer #1 (Remarks to the Author):

The authors have reviewed their manuscript thoroughly. In my opinion, this nice work can be accepted as is.

Reviewer #2 (Remarks to the Author):

Revised manuscript and comments provided by the authors are sufficient. I would recommend publication of the manuscript in the current form.